# Infants with Congenital Diseases Identified through Newborn Screening—United States, 2018–2020

**DOI:** 10.3390/ijns9020023

**Published:** 2023-04-13

**Authors:** Amy Gaviglio, Sarah McKasson, Sikha Singh, Jelili Ojodu

**Affiliations:** Association of Public Health Laboratories, Silver Spring, MD 20910, USA

**Keywords:** newborn screening, rare disease, birth prevalence

## Abstract

Newborn screening (NBS) is a state or territory-based public health system that screens newborns for congenital diseases that typically do not present with clinical symptoms at birth but can cause significant mortality and morbidity if not detected or treated quickly. NBS continues to be one of the most successful public health interventions in the US, providing early detection and intervention to all infants. The increase in overall birth prevalence of core Recommended Uniform Screening Panel (RUSP) diseases detected via dried blood spot (DBS) specimens from 2015–2017 (17.50–18.31 per 10,000) to 2018–2020 (20.07 per 10,000), as reported into the APHL NewSTEPs database, affirms the importance and impact of NBS programs. This report presents aggregate numbers and birth prevalence of diseases detected by DBS on the RUSP from 2018–2020, including data from fifty US states and two territories.

## 1. Introduction

Newborn screening (NBS) in the United States is a state or territory-based public health system that screens newborns for congenital diseases that typically do not present with clinical symptoms at birth but can cause significant mortality and morbidity if not detected or treated quickly. Newborn screening is available to all newborns (N = 3,659,289 in 2021) in the United States, regardless of location or place of birth.

The Health and Human Services (HHS) Advisory Committee on Heritable Disor-ders in Newborns and Children (ACHDNC) evaluates and recommends diseases for inclusion on the Recommended Uniform Screening Panel (RUSP). The ACHDNC continues to utilize the Wilson and Jungner criteria as a foundation for their evidentiary review in determining whether a disease is appropriate for state-mandated population screening [1]. From 2015–2020, four diseases were added to the RUSP, making a total of 35 diseases that the ACHDNC recommended screening for by NBS programs. Although the RUSP serves as a general recommendation, the implementation of screening for each disease ultimately falls to each individual NBS program, which must balance their own population characteristics, resources, and abilities when deciding whether to include a disease on their program’s universal screening panel.

Variations continue to exist in how many NBS programs screen for the most re-cent disease additions to the RUSP. As of December 2022, 34 NBS programs screen for mucopolysaccharidosis I (MPS I); 37 NBS programs screen for Pompe disease; 48 NBS programs screen for spinal muscular atrophy (SMA); and 32 NBS programs screen for X-linked adrenoleukodystrophy (X-ALD) [2]. The number of states and territories offering population screening for these diseases has increased as of 2020. Differences in the number of diseases screened in each NBS program and the methodology used for screening are due to a variety of factors, including but not limited to the birth prevalence of a particular disease across and within certain populations; the NBS programs’ resources for screening (including laboratory requirements and follow-up needs); jurisdictional public health and medical system infrastructure; legislative mandates; and costs [3,4,5].

A 2020 report estimated that approximately 6,646 babies might be identified each year with a disease screened through dried blood spot (DBS)-based newborn screening using aggregate confirmed case data entered into the Association of Public Health Laboratories’ Newborn Screening Technical assistance and Evaluation Program’s (NewSTEPs) data repository from 2015–2017 [6]. The analysis described here updates and supplements previously published data using both 2015–2017 and 2018–2020 aggregate case data collected in the NewSTEPs data repository to provide updated estimates of birth prevalence. Birth prevalence estimates are often delayed due to the time needed to confirm presumptively identified cases through newborn screening, especially for those cases with non-classic presentations.

NBS programs within the United States utilize a coordinated system of notification and reporting to help ensure that actionable screening results receive appropriate follow-up and outcome determination [7]. Despite overall advances in electronic data exchange, much of the NBS diagnostic confirmation is still accomplished through largely manual processes, often resulting in the delayed ability of NBS programs to obtain diagnostic outcomes, and/or collect enough diagnostic data to facilitate the use of public health-defined case definitions and common diagnostic classifications.

While NewSTEPs has developed and published consensus public health surveillance case definitions for diseases on the RUSP in an effort to facilitate cross-program comparisons, diagnostic data collection at this level remains complicated for the reasons mentioned above [8]. As a result, total case counts reported by NBS programs for each year are utilized rather than trying to obtain case information at an individual level with detailed demographic and diagnostic information.

## 2. Materials and Methods

### 2.1. NewSTEPs Data Repository

NewSTEPs is a program funded through a cooperative agreement between the Asso-ciation of Public Health Laboratories (APHL) and the Genetic Services Branch of the US Health Resources and Services Administration (HRSA) [9]. The activities of NewSTEPs, including maintaining a repository that captures comprehensive NBS data, are essential in ensuring that NBS programs can adequately evaluate themselves and screening outcomes using standardized performance metrics. One component of the data collected in the repository is aggregate counts of confirmed cases of core RUSP diseases.

NewSTEPs collects data on a voluntary basis from all 50 states, the District of Columbia, Guam, and Puerto Rico for a total of 53 NBS programs with data represented in the data repository. Data are collected in accordance with an established data entry timeline. Case data collected are requested in the repository from two years prior, on an annual basis, to accommodate the time it takes to resolve and close out cases.

### 2.2. Aggregate Case Data Request

To ease the data entry burden for NBS programs, NewSTEPs fielded a survey via SurveyMonkey (http://www.surveymonkey.com) to NBS personnel that had permission and access to enter case data into the NewSTEPs data repository. The survey included textboxes to enter total confirmed case counts for 2018–2020 per disease on the RUSP. Cases from 2015–2017 were updated directly by programs in the NewSTEPs data repository. Only core RUSP diseases detected by DBS specimens from 2015–2020 were included for the purposes of this report. The subtypes of sickle cell disease (SCD) listed as separate diseases on the RUSP were combined into one category of “Presence of Hb S” to mirror collection terminology in the NewSTEPs data repository. These diseases are collectively referred to as sickling hemoglobinopathies and do not include cases with an identified hemoglobinopathy trait.

Survey data were entered into the NewSTEPs data repository so that the reported data could be reviewed further and updated accordingly by newborn screening programs. The NewSTEPs program reviewed the entered data to assess for outliers or potential data integrity issues and subsequently requested that NBS programs verify aggregate case data relating to both the survey entry for 2018–2020 data as well as for historical data previously entered for 2015–2017 cases. NewSTEPs received aggregate case data from 49 states and 2 US territories. Aggregate cases (including updated case data for 2015–2017) published in this report are representative of the NewSTEPs data repository as of 21 March 2023.

Aggregate case data for 2015–2020, screening status, and implementation dates were queried from the repository using a combination of Structured Query Language (SQL) and Tableau Prep (Tableau, Seattle, WA, USA). Annual birth data, stratified by state, published by the Centers for Disease Control and Prevention (CDC) were used for 2018–2020. For each of the diseases, birth prevalence estimates for 2015–2017 and 2018–2020 were calculated using the aggregate case counts received from each jurisdiction and annual birth data from the CDC. If aggregate cases were not reported for any given year, that state or jurisdiction was excluded from the analysis. For each disease, denominator data included only births during months for which population screening was available in the state or territory and for which the NBS program reported aggregate case data to NewSTEPs. When universal screening was implemented during a particular year, a fractional analysis was performed to determine the approximate number of births receiving screening. This was conducted by dividing the annual births by 12, and then multiplying that value by the number of months in that year in which the program provided population screening.

## 3. Results

The number of programs universally screening for MPS I, Pompe disease, SMA, and X-ALD increased from 2018 to 2020 (Table 1). These four diseases were added to the RUSP from 2015 to 2018, and population screening continued to be implemented throughout the nation during the three-year timeframe presented here.

New (2018–2020) and updated (2015–2017) prevalence estimates for each of the DBS RUSP diseases were combined for all states and territories, except for Mississippi and the District of Columbia (Table 2). Not all states or territories were included for some diseases when it was determined that a state or territory was not providing universal screening for that disease. In the previously reported data from 2015–2017, the birth prevalence for DBS diseases on the RUSP was estimated to be 17.50 per 10,000. Using updated data in the NewSTEPs data repository for those years, there was a slight increase in birth prevalence to 18.31 per 10,000 for that same timeframe. From 2018–2020, the birth prevalence for DBS diseases on the RUSP increased to 20.07 per 10,000. Applying this to the number of live births in the US in 2021 (N = 3,686,219), it is expected that approximately 7389 infants will be identified through DBS-based NBS. In 2018–2020, the most prevalent DBS diseases are SCD (4.98 per 10,000), primary congenital hypothyroidism (CH) (6.69 per 10,000), and cystic fibrosis (CF) (2.29 per 10,000).

## 4. Discussion

This is a follow-up report on previously published data on the prevalence of NBS diseases in the United States. Based on live births in 2021, approximately 8180 infants with a disease on the core RUSP will now be detected annually through DBS-based NBS. Notable changes in prevalence for many of the diseases have occurred since the previous estimates based on 2015–2017 births [6].

A decrease in prevalence between 2018–2020 and the updated 2015–2017 case data was seen for several diseases, most markedly in carnitine update defect/carnitine transport defect (CUD), X-linked adrenoleukodystrophy (XALD), severe combined immune deficiency (SCID), hemoglobinopathies (HGBs), and 3-methylcrotonyl-CoA carboxylase deficiency (3-MCC). An increase in prevalence as compared to 2015–2017 was seen in numerous diseases. The largest increases occurred in congenital hypothyroidism (CH), cystic fibrosis (CF), Pompe disease, biotindiase deficiency (BIO), and PAH deficiency/phenylketonuria (PAH/PKU). This is the first report of spinal muscular atrophy (SMA) birth prevalence since its inclusion on the RUSP in 2018.

The increased prevalence of CF during 2018–2020 compared with 2015–2017 might reflect a form of surveillance bias due to the increased use of expanded molecular analysis, which has the potential to detect more inconclusive cases that are often subsequently characterized as cystic fibrosis-related metabolic syndrome (CRMS) [10]. The increase in CH prevalence might be a continuation of long-term trends related to the higher proportion of US births to Hispanic parents, who have been shown to have a higher birth prevalence of CH [11,12]. Additionally, NBS programs and clinicians vary in their screening algorithms used to detect CH (i.e., using TSH versus T4 or both analytes) as well as their follow-up and diagnostic processes. As a result, characterization of newborns with CH may fluctuate depending on how conservative programs or clinicians are in considering a child to have CH. The lack of long-term follow-up data on these newborns also limits the ability to determine how many CH cases represent permanent versus transient cases [13]. For SCD, it was previously suggested that the shifting prevalence of SCD might reflect more births to parents originating from countries where SCD is relatively common. However, the decrease in SCD for 2018–2020 might be indicative of the downward trend in international migration into the US over the past few years as a result of the COVID-19 pandemic and changing policies [14].

As with the previous publication of these data, the findings in this report are subject to several limitations. First, variations in the prevalence of individual disorders across states and territories might reflect differences in screening methods, case definitions, follow-up, and reporting practices versus a true difference in birth prevalence. Although NewSTEPs recommends uniform case definitions, it is possible that not all NBS programs applied these definitions consistently. Case counts are self-reported by each NBS program, and, although programs were asked to confirm counts, variations may still exist depending on criteria utilized within the program and the clinical knowledge of the individual(s) completing the survey.

The above leads to the second limitation, whereby programs faced challenges in classifying certain diseases, especially those with milder or later-onset phenotypes. As a result, birth prevalence may be initially overestimated as programs choose to count a case prior to obtaining clinical verification. Over time, birth prevalence and case counts may change as clinical symptoms do or do not appear and as additional disease-specific knowledge is gained. For this reason, Table 2 includes updated aggregate cases collected in NewSTEPs for 2015–2017.

While this analysis indicates that NBS programs continue to fulfill the essential role of ensuring the health and well-being of babies in the US, additional data collection and analyses are needed for more robust harmonization of how diseases on NBS panels are counted and classified, so that birth prevalence and other metrics, such as positive predictive value, can be more accurately determined. However, despite the need for refinement of data collection efforts, it is clear that NBS, coupled with the inclusion of more diseases, continues to result in life-improving intervention for more than 8500 infants and their families each year.

## Figures and Tables

**Table 1 IJNS-09-00023-t001:** Number of newborn screening programs universally screening for Pompe disease, MPS I, X-ALD, and SMA by the end of each respective calendar year.

Disease	Year Added to RUSP	2018 ^1^	2019 ^1^	2020 ^1^
Pompe disease	2015	16	21	25
MPS I	2016	15	20	24
X-ALD	2016	13	17	21
SMA	2018	5	15	29

^1^ Note that for each year, NBS programs may have implemented universal screening at different time points during the year.

**Table 2 IJNS-09-00023-t002:** National recommended uniform screening panel disease counts and birth prevalence.

Disorder	MMWR * No. of Cases Reported 2015–2017	MMWR No. of Births	MMWR Rate (Cases Per 10,000 Births)	No. of Cases Reported 2015–2017	No. of Births	Rate (Cases Per 10,000 Births)	No. of Cases Reported 2018–2020	No. of Births	Rate (Cases Per 10,000 Births)	Rate Difference
**Amino Acid Disorders**										
PAH deficiency	691	11,750,876	0.59	724	11,843,949	0.61	852	11,086,342	0.77	0.16
MSUD	64	11,750,876	0.05	65	11,843,949	0.05	56	11,086,342	0.05	0.00
Homocystinuria	18	11,750,876	0.02	19	11,843,949	0.02	16	11,086,342	0.01	0.00
Citrullinemia, type I	75	11,750,876	0.06	76	11,843,949	0.06	69	11,086,342	0.06	0.00
Argininosuccinic aciduria	59	11,750,876	0.05	59	11,753,317	0.05	57	11,086,342	0.05	0.0
Tyrosinemia, type I	22	11,750,876	0.02	22	11,669,593	0.02	39	11,025,632	0.04	0.02
**Organic Acid Disorders**										
Isovaleric acidemia	84	11,750,876	0.07	84	11,843,949	0.07	67	11,086,342	0.06	−0.01
Glutaric acidemia, type I	104	11,750,876	0.09	104	11,843,949	0.09	106	11,086,342	0.10	0.01
3-Hydroxy-3-methylglutaric aciduria	6	11,750,876	0.01	6	11,843,949	0.01	8	11,086,342	0.01	0.00
3-Methylcrotonyl-CoA carboxylase deficiency	293	11,750,876	0.25	298	11,843,949	0.25	224	11,086,342	0.20	−0.05
Methylmalonic acidemia (methylmalonyl-CoA mutase)	22	11,750,876	0.02	29	11,843,949	0.02	45	11,086,342	0.04	0.02
Propionic acidemia	63	11,750,876	0.05	63	11,843,949	0.05	62	11,086,342	0.06	0.00
Methylmalonic acidemia (cobalamin disorders)	43	11,750,876	0.04	41	11,843,949	0.03	23	11,086,342	0.02	−0.01
Holocarboxylase synthase deficiency	6	11,750,876	0.01	5	11,843,949	0.004	7	11,086,342	0.01	0.002
β-Ketothiolase deficiency	8	11,750,876	0.01	9	11,843,949	0.01	15	11,086,342	0.01	0.01
**Fatty Acid Oxidation Disorders**										
Medium-chain acyl-CoA dehydrogenase deficiency	689	11,750,876	0.59	690	11,843,949	0.58	651	11,086,342	0.59	0.00
Very long-chain acyl-CoA dehydrogenase deficiency	206	11,750,876	0.18	206	11,843,949	0.17	204	11,086,342	0.18	0.01
Long-chain L-3 hydroxyacyl-CoA dehydrogenase deficiency	26	11,750,876	0.02	27	11,843,949	0.02	32	11,086,342	0.03	0.01
Trifunctional protein deficiency	6	11,750,876	0.01	6	11,843,949	0.01	8	11,086,342	0.01	0.00
Carnitine uptake defect/carnitine transport defect	138	11,750,876	0.12	141	11,843,949	0.12	107	11,086,342	0.10	−0.02
**Hemoglobinopathies**										
SCD (includes S,S disease, S,beta-thalassemia, and S,C disease)	5808	11,750,876	4.94	6076	11,843,949	5.13	5517	11,086,342	4.98	−0.15
**Endocrine Diseases**										
Primary congenital hypothyroidism	6629	11,049,582	6	6967	11,843,949	5.88	7421	11,086,342	6.69	0.81
Congenital adrenal hyperplasia	819	11,750,876	0.7	810	11,843,949	0.68	781	11,086,342	0.70	0.02
**Lysosomal Diseases**										
Glycogen storage disease, type II (Pompe)	62	1,828,917	0.34	59	1,952,056	0.30	309	5,479,082	0.56	0.26
Mucopolysaccharidosis, type I	11	965,027	0.11	12	1,001,675	0.12	71	5,121,441	0.14	0.02
**Other DBS Screening Diseases**										
Biotinidase deficiency	477	11,750,876	0.41	488	11,843,949	0.41	655	11,086,342	0.59	0.18
Cystic fibrosis	2145	11,750,876	1.83	2451	11,735,515	2.09	2518	10,983,899	2.29	0.20
Classical galactosemia	249	11,750,876	0.21	256	11,843,949	0.22	216	11,086,342	0.19	−0.02
Severe combined immune deficiency	220	9,763,119	0.23	181	9,673,588	0.19	185	10,947,551	0.17	−0.02
Spinal muscular atrophy	NA	NA	NA	NA	NA	NA	219	3,185,560	0.69	NA
X-linked adrenoleukodystrophy	83	1,561,394	0.53	161	1,556,036	1.03	345	5,125,176	0.67	−0.36
**Total infants identified via DBS screening**	19,126		17.50	20,135		18.31	20,885		20.07	

* MMWR = Morbidity and Mortality Weekly Report.

## Data Availability

Not applicable.

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
