# Peer review of "Infants with Congenital Diseases Identified through Newborn Screening—United States, 2018–2020"

_2409-515X, 2023, doi:10.3390/ijns9020023_

Round 1

Reviewer 1 Report

This manuscript describes prevalence data for inborn errors of metabolism covered in newborn screening in the US. Such data are valuable and strongly needed for international comparisons and harmonizations. The manuscript is well written, concise but still contains the necessary information on the topic. I recommend accepting the paper after consideration of the following minor notes:

1. Please consider mentioning NewSTEPs in the Abstract.

2. Please define the abbreviation MMWR in Table 2.

Author Response

Thank you so much for this review.

Both points have been addressed in the revised manuscript. Please see the attachment.

Reviewer 2 Report

This is an interesting paper collecting information on all NBS programs in the USA. The paper is clearly written and easy to read. Some thoughts for the authors to consider that would improve the overall impression of the article;

It may be difficult to include but it would have been nice to also have PPV values for the program(s). This is indeed extremely important and if not possible to include I think the authors could say something about this at least in the discussion.

The reader of the paper is given a very positive view on adding new disorders to the panel. There have been two recent important papers/comments published in IJNS by Dr Robert Currier and Drs David Millington/Can Ficicioglu that addresses concerns and I think including something on this this could improve this work.

Why do you think that SMA (latest on the RUSP) is the most frequent among states/programs? Can you speculate on this?

It could be nice for the reader to already in the introduction have the overall population in USA as well as the annual birth rate(s).

In results and table 2 I would prefer presenting prevalence data differently but I guess this may be more of an editorial decission. 1: 570 on line 117 or 1: 17 000 for PAH def. would be easier for international comparison (I think).

Is it possible to also include 2021 and 2022 in table 1?

Screening algorithms in the NBS programs could also be mentioned in the discusssion. For CH for example (line 144-151), this is probably one of the most important factors behind the different prevalences seen.

Author Response

Thank you so very much for the thorough review. Please see below for responses to your comments and the revised manuscript attached.

1) Reviewer Comment: It may be difficult to include but it would have been nice to also have PPV values for the program(s). This is indeed extremely important and if not possible to include I think the authors could say something about this at least in the discussion.

Author Response: We agree that this would be nice to include; however, it is not captured in an easily comparable way within NewSTEPs. We have added this into the discussion as a consideration for future needs. We will also certainly work within our membership to encourage them to publish this information.

2) Reviewer Comment: The reader of the paper is given a very positive view on adding new disorders to the panel. There have been two recent important papers/comments published in IJNS by Dr Robert Currier and Drs David Millington/Can Ficicioglu that addresses concerns and I think including something on this this could improve this work.

Author Response: The intent of this paper is not as a commentary on the ease or burdens of state NBS panel expansion, rather a statement of current data available re: birth prevalence. We are not intending to speculate on considerations for addition to panels.

3) Reviewer Comment: Why do you think that SMA (latest on the RUSP) is the most frequent among states/programs? Can you speculate on this?

Author Response: This is not within the scope of this particular paper, however, this information is being published within IJNS in a separate paper very shortly by NewSTEPs and partners. However, this is likely due to the ability to multiplex this disease easily with an existing assay as well as more clear effectiveness of treatment.

4) Reviewer Comment: It could be nice for the reader to already in the introduction have the overall population in USA as well as the annual birth rate(s).

Author Response: This has been added.

5) Reviewer Comment: In results and table 2 I would prefer presenting prevalence data differently but I guess this may be more of an editorial decission. 1: 570 on line 117 or 1: 17 000 for PAH def. would be easier for international comparison (I think).

Author Response: This was an editorial decision in order to mirror the previous publication on this data that was published in MMWR.

6) Reviewer Comment Is it possible to also include 2021 and 2022 in table 1?

Author Response: This data is not yet available as it can take time for confirmation of cases. We are trying to determine if there are ways to publish this data in a more timely manner in the future while allowing due time for positive cases to resolve.

7) Reviewer Comment: Screening algorithms in the NBS programs could also be mentioned in the discusssion. For CH for example (line 144-151), this is probably one of the most important factors behind the different prevalences seen.

Author Response. This is an excellent point. Consideration around the screening algorithm used as an explanation has been added in the discussion.
